# Change in unemployment by social vulnerability among United States counties with rapid increases in COVID-19 incidence—July 1–October 31, 2020

**Shichao Tang** *, **Libby Horter**, **Karin Bosh, Ahmed M. Kassem, Emily B. Kahn, Jessica N. Ricaldi**, **Leah Zilversmit Pao, Gloria J. Kang, Christa-Marie Singleton, Tiebin Liu, Isabel Thomas, Carol Y. Rao**

COVID-19 Response Team, Centers for Disease Control and Prevention, Atlanta, Georgia, United States of America

* stang2@cdc.gov

**Data Availability Statement:** The data is publicly available as described in the manuscript. The US COVID-19 case data is from USAFacts: https://

## Abstract

### Objective

During the COVID-19 pandemic, the unemployment rate in the United States peaked at 14.8% in April 2020. We examined patterns in unemployment following this peak in counties with rapid increases in COVID-19 incidence.

### Method

We used CDC aggregate county data to identify counties with rapid increases in COVID-19 incidence (rapid riser counties) during July 1–October 31, 2020. We used a linear regression model with fixed effect to calculate the change of unemployment rate difference in these counties, stratified by the county's social vulnerability (an indicator compiled by CDC) in the two months before the rapid riser index month compared to the index month plus one month after the index month.

### Results

Among the 585 (19% of U.S. counties) rapid riser counties identified, the unemployment rate gap between the most and least socially vulnerable counties widened by 0.40 percentage point (p<0.01) after experiencing a rapid rise in COVID-19 incidence. Driving the gap were counties with lower socioeconomic status, with a higher percentage of people in racial and ethnic minority groups, and with limited English proficiency.

### Conclusion

The widened unemployment gap after COVID-19 incidence rapid rise between the most and least socially vulnerable counties suggests that it may take longer for socially and economically disadvantaged communities to recover. Loss of income and benefits due to unemployment could hinder behaviors that prevent spread of COVID-19 (e.g., seeking

usafacts.org/visualizations/coronavirus-covid-19-spread-map/ The Social Vulnerability Index data is from Centers for Disease Control and Prevention: https://www.atsdr.cdc.gov/placeandhealth/svi/data_documentation_download.html The County population estimate (2019) data is from the U.S. Census Bureau: https://www.census.gov/data/datasets/time-series/demo/popest/2010s-counties-total.html The US county-level unemployment statistics data is from the U.S. Bureau of Labor Statistics: https://www.bls.gov/lau/#tables".

**Funding:** The authors received no specific funding for this work.

**Competing interests:** The authors have declared that no competing interests exist.

healthcare) and could impede response efforts including testing and vaccination. Addressing the social needs within these vulnerable communities could help support public health response measures.

## Introduction

In the United States, the COVID-19 pandemic has resulted in more than 62,000,000 reported cases and more than 840,000 associated deaths as of January 13, 2022 [1]. Although the pandemic may have affected most people living in the United States in some way, the impact has not been equal across communities [2, 3]. Societal factors such as poverty, lack of access to transportation, crowded households, racial and ethnic inequalities, work-related hardship or risk (e.g., unemployment, underemployment, and distribution of essential and/or public facing jobs), and other social conditions, affect a community's ability to cope with a disaster [4] like the COVID-19 pandemic. Social vulnerability is a term that refers to the potential negative effects on communities caused by stresses like disease outbreaks [5]. Counties that are more socially vulnerable are more likely to have high COVID-19 incidence [3]. For example, racial and ethnic minority groups have had disproportionately high numbers of COVID-19 cases and associated hospitalizations and deaths [6–8]. Counties with a higher percentage of people living in crowded housing are more likely to experience rapidly increasing COVID-19 incidence [3].

In addition to the direct negative health impacts of COVID-19, the pandemic also has impacted the economy: the unemployment rate peaked at 14.8% in April 2020 when more than 6,000,000 people filed initial claims of unemployment insurance in a week [9]. Persons from racial and ethnic minority groups are overrepresented in the services industries such as leisure and hospitality that were disproportionally impacted by the economic downturn caused by COVID-19 [10–12]. Rapid increase of COVID-19 incidence in a community may lead to business closures and worker layoffs, and the impact on socially vulnerable communities may be greater than less vulnerable communities [13].

In this study, we sought to examine changes in unemployment rates among counties with rapid increases in COVID-19 incidence (rapid riser counties). Since March 8, 2020, the Centers for Disease Control and Prevention (CDC) has used county-level case counts and standard criteria to identify counties with rapidly increasing COVID-19 incidence, known as rapid riser counties. These criteria reflect a rapid increase of COVID-19 incidence within a short period of time as a method to focus public health efforts in these communities with disproportionately high COVID-19 rates [14]. Specifically, we described unemployment changes in rapid riser counties by the CDC social vulnerability index (SVI), an index that measures the potential negative effects on communities caused by external stresses on human health and helps local officials identify socially vulnerable communities that may need support before, during, or after disasters [5]. The SVI has been used in prior county-level COVID-19 studies [3, 15].

## Materials and methods

Using CDC aggregate county data (a primary case reporting feed utilized for federal response and managed by CDC) of reported daily COVID-19 case counts during July 1–October 31, 2020, we identified rapid riser counties, defined as those that met all of the following standardized criteria [14] on the day assessed: 1) >100 new cases in the most recent 7 days, 2) >0% change in 7-day incidence, 3) a decrease of <60% or an increase in the most recent 3-day

 

incidence over the preceding 3-day incidence, and 4) a 7-day incidence/30-day incidence ratio >0.31 of COVID-19; and, met one or both of the following triggering criteria: 1) >60% change in the most recent 3-day incidence, or 2) >60% change in the most recent 7-day incidence of COVID-19. These criteria were developed through a collaborative process involving multiple federal agencies [14]. For this analysis, we categorized a county as a rapid riser if the county met the standardized daily rapid riser criteria on at least three days in the week; we chose this approach in order to minimize misclassification due to reporting errors (e.g., as might occur if a county reported COVID-19 case counts in multi-day batches instead of daily). We defined the index month based on the month when a county first experienced a rapid rise in COVID-19 incidence; and employed a mid-month cut-point where, if the first rapid riser alert was on or after the 15th day of the month, the next month was assigned as the index month. Pre-index months were defined as the 2 months preceding the index month. Post-index months were defined as the index month plus the following month.

We extracted county-level SVI data from the most recent database, CDC/ATSDR SVI database 2018 [5]. The overall SVI and the four SVI themes were used: 1) socioeconomic status including "below poverty", "unemployed", "income", and "no high school diploma"; 2) household composition & disability including "aged 65 years or older", "aged 17 years or younger", "civilian with a disability", and "single-parent households"; 3) minority status & languages including "minority" and "speak English 'less than well'"; and 4) housing type & transportation including "multi-unit structures", "mobile homes", "crowding", "no vehicle", and "group quarters". Both overall SVI and four SVI themes for each of the counties included were categorized by quartiles based on SVI and the SVI themes in the 3,141 US counties with complete data (99.9% of US counties or county equivalents). One U.S. county was excluded due to missing SVI data. The fourth quartile represents the most socially vulnerable U.S. counties, and the first quartile represents the least socially vulnerable U.S. counties.

We obtained monthly unemployment rates at the county level from January through November 2020 from the U.S. Bureau of Labor Statistics (BLS) [16]. BLS defines unemployment rate as the percent of unemployed persons of the civilian labor force [17]. The county-level unemployment rate gap was defined as the rate difference comparing first SVI quartile (referent) to other quartiles, overall and by SVI themes. We selected the July–October 2020 timeframe to identify rapid riser counties to understand the impact of county-level COVID-19 rapid rise on unemployment rates after the initial phase of the COVID-19 pandemic. For each rapid riser county, we examined unemployment rates within a period of four months (i.e., two months before an index month, the index month, and the following month). We calculated the difference of the average unemployment rate gap between pre-index months and post-index months, using a linear regression model (Eq 1). Five separate regression models—one by overall SVI and four SVI themes were estimated. The coefficients of the interaction terms for each SVI quartile represent the changes in the unemployment rate gap. The standard error was clustered at the county level while controlling for month- and county-fixed effects as well as the number of days a county met rapid riser criteria within the 60 days after counties were first flagged as a rapid riser. The statistical analysis was performed using Stata 15 (College Station, TX).

$$Unemployment\ Rate_{ct} = \alpha + \beta_1 SVIQ2_c + \beta_2 SVIQ3_c + \beta_3 SVIQ4_c + \beta_4 Exposure_{ct} + \beta_5 SVIQ2_c \times Exposure_{ct} + \beta_6 SVIQ3_c \times Exposure_{ct} + \beta_7 SVIQ4_c \times Exposure_{ct} + \theta RR\ intensity_{ct} + \gamma County_c + \varphi Month_t + \varepsilon_{ct}$$

(1)

Where $Unemployment\ Rate_{ct}$ represents the unemployment rate of county $c$ during time $t$, $SVIQ2_c$ $SVIQ3_c$, and $SVIQ4_c$ indicates the status of the second, third, and fourth quartile of the SVI for county $c$, $Exposure_{ct}$ is the rapid rise status for county $c$ during time $t$, $RR\ intensity_{ct}$, is

the number of days a county met rapid riser criteria within 60 days after counties were first flagged as rapid riser for county $c$ during time $t$, $Country_c$ and $Month_t$ represent county- and month- fixed effects.

Since the unemployment rate appears in both side of the equation (The overall SVI and the SVI socioeconomic status theme include unemployment rate), sensitivity analyses by excluding unemployment from the overall SVI and SVI socioeconomic status theme were conducted to see if this would significantly impact the results.

This activity was reviewed by CDC and was conducted consistent with applicable federal law and CDC policy. 45 C.F.R. part 46, 21 C.F.R. part 56; 42 U.S.C. Sect. 241(d); 5 U.S.C. Sect. 552a; 44 U.S.C. Sect. 3501 et seq.

## Results

From July 1 through October 31, 2020, 585 (19%) of 3,141 U.S. counties were considered a COVID-19 rapid riser county for the first time (S1 Fig): 243 (42%) in July, 134 (23%) in August, 73 (12%) in September, and 135 (23%) in October. Among them, 29% were in the Midwest region, 2% were in the Northeast region, 58% were in the South region, and 11% were in the West region. Forty seven percent of those rapid riser counties are from metropolitan areas, and 53% of them are from non-metropolitan areas. Regarding unemployment rates among these 585 counties from January through November 2020, average unemployment rates peaked in April and then started to decrease (Fig 1). There was an unemployment rate gap between the most socially vulnerable counties and least socially vulnerable counties before March 2020. The gap narrowed in April when the unemployment rate reached its highest level and widened again since July 2020.

Average unemployment rate in the rapid riser counties changed over the time period examined in the analysis by overall SVI quartiles (Fig 2). The unemployment rate gap widened by 0.40 percentage points (95% CI: 0.15%, 0.65%) between the most and the least socially vulnerable counties when comparing rates between pre-index months and post-index months (Table 1). Comparing the most socially vulnerable counties to the least, for the socioeconomic status SVI theme, the unemployment rate gap widened by 0.36 percentage point (95% CI: 0.11%, 0.61%); and for the racial and ethnic minority and English proficiency SVI theme, the unemployment rate gap widened by 0.38 percentage point (95% CI: 0.06%, 0.70%) between the most and the least socially vulnerable counties. No significant changes were found with respect to household composition and disability SVI theme or the housing type and transportation SVI theme. The sensitivity analyses results are not significantly different from the results from our main analysis and the conclusion remains the same (S1 Table).

## Discussion

In U.S. counties experiencing rapid increases in COVID-19 incidence after the initial unemployment peak in April, unemployment rate gaps widened between the most and the least socially vulnerable counties when comparing rates between pre-index months and post-index months. The average unemployment rate gap between the most and the least socially vulnerable counties before the index month was 2.64%. An unemployment gap of 0.40 percentage point represents about a 15% increase. By the SVI themes, the unemployment rate gap widened significantly for counties with the lowest socioeconomic status and highest proportion of racial and ethnic minority residents and lowest English proficiency. These findings show that the already existing unemployment gap worsened when COVID-19 rapid rise occurred in communities.

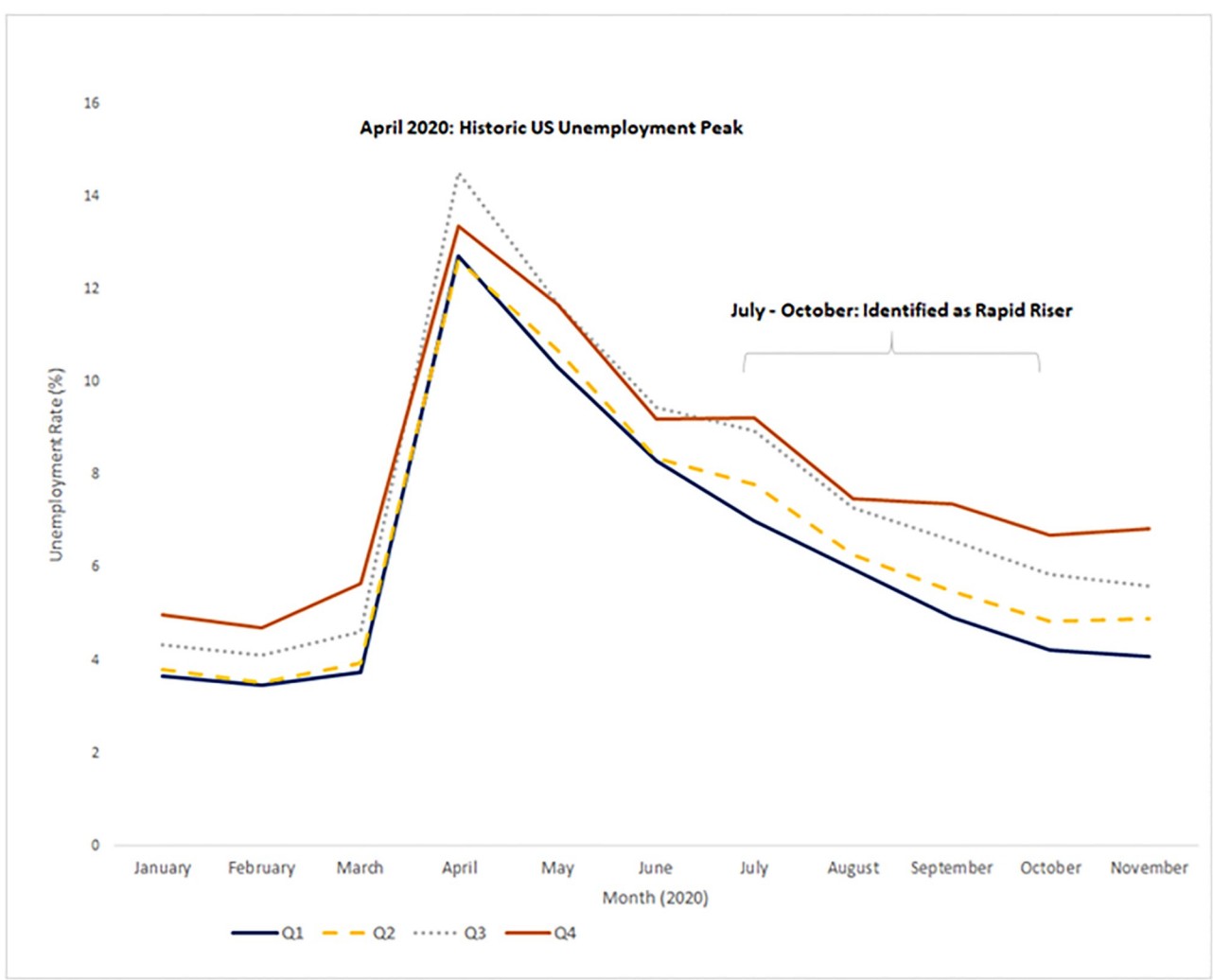

**Fig 1. Monthly unemployment rate by Social Vulnerability Index (SVI) quartile in 585 U.S. counties identified as a rapid riser in COVID-19 incidence[†] - - - United States, January-November, 2020.** †Rapid riser counties were defined as those that met all of the following criteria: 1) >100 new cases in recent week, 2) >0% change in the 7-day incidence, 3) >-60% change in the 3-day incidence, and 4) a 7-day incidence / 30-day incidence ratio >0.31. In addition, rapid riser counties met one or both of the following triggering criteria: 1) >60% change in 3-day incidence, or 2) >60% change in 7-day incidence. For this analysis, we categorized a county as a rapid riser if the county met the standardized daily rapid riser criteria on at least three days in the week.

However, we did not find significant changes for the household composition & disability theme and housing type & transportation theme, which suggests that these two SVI themes may be less relevant to the widening of the unemployment gap after a county experienced the COVID-19 incidence rapid rise compared to the other two SVI themes.

Previously published research showed that communities with higher social vulnerability are more likely to become rapid risers of COVID-19 incidence [3]. Our findings suggest that after-effects of rises in incidence in these counties can include adverse economic impacts on the respective communities. The widened unemployment gap after COVID-19 incidence rapid rise between the most and least socially vulnerable counties suggests that it may take longer for socially and economically disadvantaged communities to recover their economy. Our findings corroborate previous studies that people in racial and ethnic minority groups and with lower income were disproportionally affected economically by COVID-19 [12, 18].

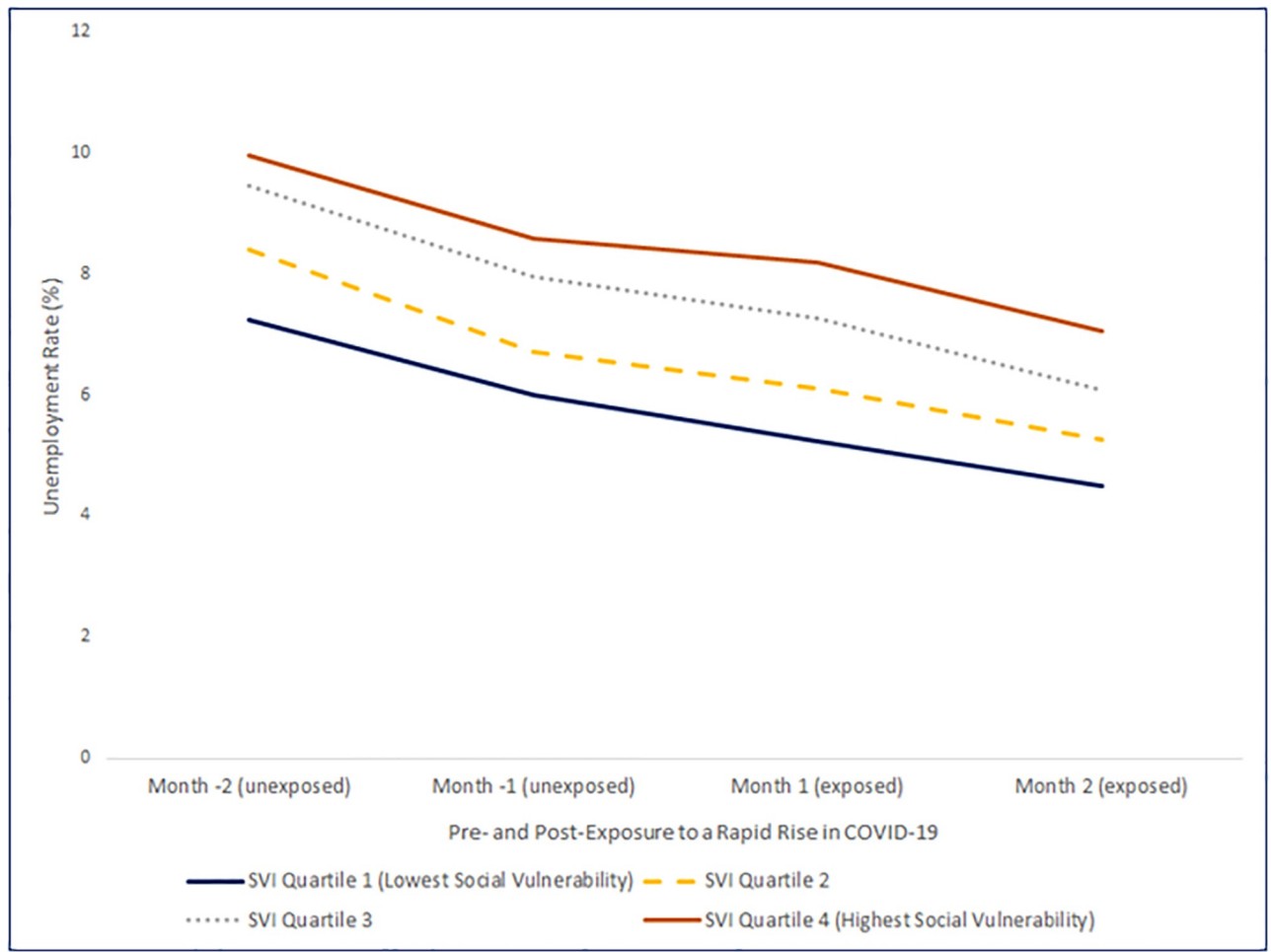

**Fig 2. County-level unemployment rate before and after[†] a rapid rise in COVID-19 incidence[§], by Social Vulnerability Index (SVI) quartiles - - - United States.** [†] Before rapid rise is defined as the 2 months preceding the rapid rise index month. After rapid rise is defined as the rapid rise index month plus the following month. [§] Rapid riser counties were defined as those that met all of the following criteria: 1) >100 new cases in recent week, 2) >0% change in the 7-day incidence, 3) >-60% change in the 3-day incidence, and 4) a 7-day incidence / 30-day incidence ratio >0.31. In addition, rapid riser counties met one or both of the following triggering criteria: 1) >60% change in 3-day incidence, or 2) >60% change in 7-day incidence. For this analysis, we categorized a county as a rapid riser if the county met the standardized daily rapid riser criteria on at least three days in the week.

This analysis is subject to at least six limitations. First, counties with smaller populations may be less likely to meet the rapid riser criteria; thus, this analysis may not be representative of less densely populated counties. Second, because we limited the analysis to first-time rapid riser counties that occurred in a time period after the U.S. unemployment peak in April 2020, we did not assess the effects in counties that were identified as rapid risers prior to July 2020 when many large urban areas were first identified as rapid riser. Almost two-thirds of US counties are non-metropolitan and a little over fifty percent of the counties examined in our sample are non-metropolitan counties. Thus, these results may not be representative of the entire United States. Third, the unemployment rate may not fully match with daily rapid rises in COVID-19 incidence since unemployment rate was reported on a monthly basis. Fourth, other policies or external shocks may happen around the same time as the COVID-19 incidence rapid rise occurred, so our estimate may be subject to bias. Fifth, the SVI does not cover every aspect of the vulnerability of a community. For instance, it does not reflect the nature of

**Table 1. Unemployment rate gap changes (β) by Social Vulnerability Index (SVI), overall and by svi theme, among rapid riser counties[†] (N = 585) before and after[¶] a rapid rise in COVID-19 incidence - - - United States.**

| Social Vulnerability Index (SVI) Quartile, by SVI Theme | Unemployment Rate Gap Change | | Number of counties |
|---|---|---|---|
| | β[§] | (95% CI) | |
| **Overall SVI** | | | |
| Q1 (lowest vulnerability) | Reference | | 96 |
| Q2 | -0.04 | (-0.29, 0.20) | 137 |
| Q3 | -0.12 | (-0.39, 0.14) | 156 |
| Q4 (highest vulnerability) | **0.40**** | (0.15, 0.65) | 196 |
| **SVI socioeconomic status theme** | | | |
| Q1(lowest vulnerability) | Reference | | 115 |
| Q2 | -0.001 | (-0.26, 0.26) | 132 |
| Q3 | 0.22 | (-0.02, 0.46) | 175 |
| Q4 (highest vulnerability) | **0.36**** | (0.11, 0.61) | 163 |
| **SVI household composition & disability theme** | | | |
| Q1(lowest vulnerability) | Reference | | 133 |
| Q2 | -0.06 | (-0.31, 0.18) | 132 |
| Q3 | 0.03 | (-0.24, 0.29) | 151 |
| Q4 (highest vulnerability) | 0.15 | (-0.11, 0.41) | 169 |
| **SVI minority status & language theme** | | | |
| Q1(lowest vulnerability) | Reference | | 72 |
| Q2 | 0.24 | (-0.02,0.51) | 172 |
| Q3 | **0.41**** | (0.14,0.68) | 197 |
| Q4 (highest vulnerability) | **0.38*** | (0.06,0.70) | 144 |
| **SVI housing type & transportation theme** | | | |
| Q1(lowest vulnerability) | Reference | | 68 |
| Q2 | -0.16 | (-0.46, 0.13) | 131 |
| Q3 | 0.01 | (-0.27, 0.29) | 185 |
| Q4 (highest vulnerability) | 0.06 | (-0.23, 0.34) | 201 |

Boldface indicates statistical significance (*$p<0.05$, **$p<0.01$).

[†]Rapid riser counties were defined as those that met all of the following criteria: 1) >100 new cases in recent week, 2) >0% change in the 7-day incidence, 3) >-60% change in the 3-day incidence, and 4) a 7-day incidence / 30-day incidence ratio >0.31. In addition, rapid riser counties met one or both of the following triggering criteria: 1) >60% change in 3-day incidence, or 2) >60% change in 7-day incidence. For this analysis, we categorized a county as a rapid riser if the county met the standardized daily rapid riser criteria on at least three days in the week.

[¶] Before rapid rise is defined as the 2 months preceding the index rapid rise month. After rapid rise is defined as the index month of rapid rise plus the following month.

[§] The coefficient of regression.

work in a community such as the proportion of essential and/or public facing jobs. Finally, the SVI is an indicator of the socio-economic conditions of a county in 2018 and it is possible that those conditions may have changed since 2018, so our findings may not reflect those potential changes. In addition, the unemployment rate appears in both sides of the regression equation, which may lead to bias of the estimate. We re-estimated our results by excluding the unemployment rate from the SVI, and the results are similar to those from the main analysis.

The findings of this study underscore the importance of socioeconomic inequality in a public health crisis such as during the COVID-19 pandemic. Existing socioeconomic inequality is associated with the risk for a county to become a rapid riser in COVID-19 incidence [3], and also increases the challenge for more socially vulnerable counties to recover from any economic downturns occurring during the pandemic. Loss of income and of benefits due to

unemployment could also hinder behaviors that slow or prevent spread of COVID-19. For instance, unemployed individuals would have less access to health care and are less likely to receive needed medical care [19, 20], which would impede response efforts including testing and vaccination [20, 21]. In socially vulnerable communities disproportionately affected by COVID-19 and experiencing high rates of unemployment, such as those with high proportion of racial and ethnic minority groups and non-English speakers residents, addressing the social needs (e.g., access to healthcare [19], food [22], health insurance [23], non-English language resources for economic benefits, health care, and accurate COVID-19 information [24, 25]) to help support public health measures such as testing, contact tracing, and vaccination may be needed during any infectious disease pandemic or public health crisis.

## Supporting information

**S1 Fig. This file contains an additional supplementary figure.** US rapid riser county map. The map was generated using SAS 9.4; basemap: UNITED STATES—COUNTIES: Copyright (C) 1996. SAS Institute Inc. Created and last modified 06/25/2015. The data displayed in the map are available at: Centers for Disease Control and Prevention/ Agency for Toxic Substances and Disease Registry/ Geospatial Research, Analysis, and Services Program. CDC/ATSDR Social Vulnerability Index [2018] Database [US]. https://www.atsdr.cdc.gov/placeandhealth/ svi/data_documentation_download.html Accessed on [January 10th, 2022]. https://usafacts. org/visualizations/coronavirus-covid-19-spread-map.
(DOCX)

**S1 Table. This file contains an additional supplementary table.** Sensitivity analysis table.
(DOCX)

## Acknowledgments

**Disclaimer**: The findings and conclusions in this report are those of the authors and do not necessarily represent the official position of the Centers for Disease Control and Prevention.

## Author Contributions

**Conceptualization:** Shichao Tang.

**Data curation:** Libby Horter.

**Formal analysis:** Shichao Tang, Libby Horter.

**Investigation:** Shichao Tang.

**Methodology:** Shichao Tang.

**Visualization:** Libby Horter, Tiebin Liu.

**Writing – original draft:** Shichao Tang.

**Writing – review & editing:** Shichao Tang, Libby Horter, Karin Bosh, Ahmed M. Kassem, Emily B. Kahn, Jessica N. Ricaldi, Leah Zilversmit Pao, Gloria J. Kang, Christa-Marie Singleton, Tiebin Liu, Isabel Thomas, Carol Y. Rao.

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
