## [Decision Letter · Decision Letter 0]

1 Dec 2021

PONE-D-21-27081Change in Unemployment by Social Vulnerability among United States Counties with Rapid Increases in COVID-19 Incidence — July 1–October 31, 2020PLOS ONE

Dear Dr. Tang,

Thank you for submitting your manuscript to PLOS ONE. After careful consideration, we feel that it has merit but does not fully meet PLOS ONE’s publication criteria as it currently stands. Therefore, we invite you to submit a revised version of the manuscript that addresses the points raised during the review process by our reviewers.

We look forward to receiving your revised manuscript.

Kind regards,

Candace C. Nelson, ScD

Academic Editor

PLOS ONE

Journal Requirements:

Reviewers' comments:

Reviewer's Responses to Questions

**Comments to the Author**

1. Is the manuscript technically sound, and do the data support the conclusions?

Reviewer #1: Yes

Reviewer #2: Yes

2. Has the statistical analysis been performed appropriately and rigorously? 

Reviewer #1: Yes

Reviewer #2: Yes

3. Have the authors made all data underlying the findings in their manuscript fully available?

Reviewer #1: Yes

Reviewer #2: Yes

4. Is the manuscript presented in an intelligible fashion and written in standard English?

Reviewer #1: Yes

Reviewer #2: Yes

5. Review Comments to the Author

Reviewer #1: I thank the authors for an interesting, important, and timely article. I have included here a few comments and questions that I think will help improve the article.

Introduction

1. Might be worth updating the introduction sentence based on more recent COVID numbers (could update this closer to publication timeline)

2. In the third sentence on the way that a community can cope with a disaster, it’s not just unemployment – perhaps rephrase to talk about other work-related factors: underemployment, distribution of essential and/or public facing jobs (which in turn put different groups of people at higher risk for COVID), as well as unemployment.

3. Unemployment is also tied to negative health impacts. Perhaps rephrase the first sentence of the second paragraph to get at the direct health impacts of COVID as well as the indirect (which impact employment, which in turn impact health � e.g. someone has to leave their work, and their work is how they received health insurance so now they are uninsured and management of chronic health conditions deteriorates, as well as the impact of being unemployed on stress levels which in turn impacts health)

4. It would help to say a little more about the SVI in the introduction. Perhaps a few additional references of how it’s been used?

Methods

5. You use SVI data from 2018 – is this the most recently available? Can you comment (or maybe you do in the discussion) about how this may differ from the situation in 2020? (I see this commented on in the limitations in the discussion, but perhaps acknowledging this time gap in the methods too?

6. You mention that there were 3.141 US counties with complete data – how many counties total in the US? What is the %complete?

Discussion

7. Can you comment on you might hypothesize how things might differ in other periods? For example, during the winter peak, initial peak, or once the vaccine became widely available?

8. One thing that I don’t think is reflected in the SVI is that not all jobs are created equal. That is, especially in times of COVID, some jobs put people at increased risk due to the nature of the work. For example, public facing jobs, essential jobs – grocery stores, bus drivers, etc. – these people are deemed working, but face increased risk compared to people that can work from home. We see racial and ethnic inequities based on who works in these jobs (see for example https://www.mass.gov/doc/characterizing-ma-workers-in-select-covid-19-essential-services-food-stores-and-urban-transit/download. Perhaps worth mentioning something about this limitation of the SVI?

Reviewer #2: This is a thoughtful and interesting analysis, examining factors associated with unemployment for rapid COVID uptake counties in the US. Thank you for examining this issue.

I was struck by potential differences between rural and urban counties and among different US regions - these potentially important differences are not currently reflected in this analysis. If possible, I would suggest additional analyses to explore these results in data subgroups (urban and rural counties; different regions). Furthermore a map of US counties color-coded to share descriptive results (eg. only high riser counties shaded; outlining from bold to thin to reflect levels of increase for unemployment; color shading for significant results - Q4 SVI SES, Q3/Q4 minority+language) could be very useful in helping local and/or state decision-makers. I also acknowledge that a figure like this could be part of additional manuscripts exploring this issue.

For the Discussion, I think it’s quite crucial to emphasize the increased need for non-English language resources in terms of unemployment benefits, employment opportunities, general healthcare, and COVID information in particular considering the impact on non-English language speakers reflected here.

I saw just a few minor typos. Please reread for copy editing.

Overall an excellent study. Thank you!

6. PLOS authors have the option to publish the peer review history of their article (what does this mean?). If published, this will include your full peer review and any attached files.

Reviewer #1: **Yes: **Emily Sparer-Fine

Reviewer #2: **Yes: **Cati Brown-Johnson

---

## [Author Response · Author response to Decision Letter 0]

4 Feb 2022

Revisions and Response to Reviewer Comments on

“Change in Unemployment by Social Vulnerability among United States Counties with Rapid Increases in COVID-19 Incidence — July 1–October 31, 2020”

(Manuscript # PONE-D-21-27081)

Dear Dr. Nelson, 

Thank you for the comments and careful review of our manuscript. We appreciate the suggestions to improve it based on yours and the reviewers’ feedback. We believe the comments from you and the reviewers have helped to strengthen our work and we describe how we addressed each below. We have carefully considered all of the comments and made the corresponding revisions. Revisions in the manuscript were highlighted in yellow. 

Reviewer #1: 

1. I thank the authors for an interesting, important, and timely article. I have included here a few comments and questions that I think will help improve the article.

Thank you for your positive comments! 

2. Introduction

a. Might be worth updating the introduction sentence based on more recent COVID numbers (could update this closer to publication timeline)

Thank you for your suggestion! We have updated with COVID numbers from January 13, 2022(Lines 69-70) and can update again when closer to publication.

“In the United States, the COVID-19 pandemic has resulted in more than 62,000,000 reported cases and more than 840,000 associated deaths as of January 13, 2022”

b. In the third sentence on the way that a community can cope with a disaster, it’s not just unemployment – perhaps rephrase to talk about other work-related factors: underemployment, distribution of essential and/or public facing jobs (which in turn put different groups of people at higher risk for COVID), as well as unemployment.

Thank you for your suggestion! We revised the sentence to include other work-related factors (Lines 73-74): Now it reads as below:

“Societal factors such as poverty, lack of access to transportation, crowded households, racial and ethnic inequalities, work-related hardship or risk (e.g., unemployment, underemployment, and distribution of essential and/or public facing jobs), and other social conditions, affect a community’s ability to cope with a disaster like the COVID-19 pandemic”

c. Unemployment is also tied to negative health impacts. Perhaps rephrase the first sentence of the second paragraph to get at the direct health impacts of COVID as well as the indirect (which impact employment, which in turn impact health � e.g., someone has to leave their work, and their work is how they received health insurance so now they are uninsured and management of chronic health conditions deteriorates, as well as the impact of being unemployed on stress levels which in turn impacts health)

Thank you for your comments and suggestions! We agree that there are direct and indirect negative health impacts of COVID-19. Our objective with the first sentence of the second paragraph was to summarize the direct negative health impacts of COVID-19 discussed in the previous paragraph to bridge to the second paragraph that relates unemployment to the impact on the economy. We included the indirect negative health impact of COVID-19 through unemployment in the discussion section in lines 236 to 242. 

For clarity, we added the word “direct” to this sentence (Line 82). Now it reads as below:

“In addition to the direct negative health impacts of COVID-19”

d. It would help to say a little more about the SVI in the introduction. Perhaps a few additional references of how it’s been used?

Thank you for your suggestion! We revised the last sentence of third paragraph to include a brief description about SVI and added a few additional references in lines 96 to 98(#3, #5 and #15). Now it reads as below:

“Specifically, we described unemployment changes in rapid riser counties by the CDC social vulnerability index (SVI), an index that measures the potential negative effects on communities caused by external stresses on human health and helps local officials identify socially vulnerable communities that may need support before, during, or after disasters (5). The SVI has been used in prior county-level COVID-19 studies (3, 15).”

(3) Dasgupta S, Bowen VB, Leidner A, Fletcher K, Musial T, Rose C, et al. Association Between Social Vulnerability and a County's Risk for Becoming a COVID-19 Hotspot - United States, June 1-July 25, 2020. MMWR Morb Mortal Wkly Rep. 2020;69(42):1535-41.

(5) Centers for Disease Control and Prevention. CDC Social Vulnerability Index 2021 [Available from: https://www.atsdr.cdc.gov/placeandhealth/svi/index.html.

(15) Hughes MM, Wang A, Grossman MK, Pun E, Whiteman A, Deng L, et al. County-Level COVID-19 Vaccination Coverage and Social Vulnerability - United States, December 14, 2020-March 1, 2021. MMWR Morb Mortal Wkly Rep. 2021;70(12):431-6.

3. Methods

a. You use SVI data from 2018 – is this the most recently available? Can you comment (or maybe you do in the discussion) about how this may differ from the situation in 2020? (I see this commented on in the limitations in the discussion, but perhaps acknowledging this time gap in the methods too?

Thank you for raising the questions! The SVI is periodically updated; the 2018 data are the most recent. We revised the method section to indicate that this is the most recently available database (Line 118). Now it reads as below:

“We extracted county-level SVI data from the most recent database, CDC/ATSDR SVI database 2018.”

b. You mention that there were 3.141 US counties with complete data – how many counties total in the US? What is the %complete?

Thank you for your question. There are 3,142 counties (or county equivalents) in the US. The SVI data accounts for 99.9% of the US counties. We revised the sentence in the Methods section (Line 126). Now it reads as below:

“Both overall SVI and four SVI themes for each of the counties included were categorized by quartiles based on SVI and the SVI themes in the 3,141 US counties with complete data (99.9% of US counties or county equivalents).”

4. Discussion

a. Can you comment on you might hypothesize how things might differ in other periods? For example, during the winter peak, initial peak, or once the vaccine became widely available?

Thank you for your comment! We agree that different times during the pandemic (e.g., winter peak, post-vaccine) would likely have different results. We cannot hypothesize how these different situations might affect the results. We think this is an interesting idea and may consider a follow up analysis and manuscript.

b. One thing that I don’t think is reflected in the SVI is that not all jobs are created equal. That is, especially in times of COVID, some jobs put people at increased risk due to the nature of the work. For example, public facing jobs, essential jobs – grocery stores, bus drivers, etc. – these people are deemed working, but face increased risk compared to people that can work from home. We see racial and ethnic inequities based on who works in these jobs (see for example https://www.mass.gov/doc/characterizing-ma-workers-in-select-covid-19-essential-services-food-stores-and-urban-transit/download. Perhaps worth mentioning something about this limitation of the SVI?

Thank you for your comment and appreciate your perspective! We agree that SVI does not cover every aspect of the vulnerability of a community. We added this to the limitation section (Lines 227-229). Now it reads:

“Fifth, the SVI does not cover every aspect of the vulnerability of a community. For instance, it does not reflect the nature of work in a community such as the proportion of essential and/or public facing jobs.”

Reviewer #2: 

1. This is a thoughtful and interesting analysis, examining factors associated with unemployment for rapid COVID uptake counties in the US. Thank you for examining this issue.

Thank you for your positive comments! 

2. I was struck by potential differences between rural and urban counties and among different US regions - these potentially important differences are not currently reflected in this analysis. If possible, I would suggest additional analyses to explore these results in data subgroups (urban and rural counties; different regions). Furthermore a map of US counties color-coded to share descriptive results (eg. only high riser counties shaded; outlining from bold to thin to reflect levels of increase for unemployment; color shading for significant results - Q4 SVI SES, Q3/Q4 minority+language) could be very useful in helping local and/or state decision-makers. I also acknowledge that a figure like this could be part of additional manuscripts exploring this issue.

Thank you for your comments and suggestions! We agree that there may be potential differences between rural and urban counties. For this analysis, however, we only examined the counties that firstly become the rapid riser during July 1-October 31, 2020 in this manuscript. We selected this timeframe to understand the impact of county-level COVID-19 rapid rise on unemployment rates after the initial phase of the COVID-19 pandemic in April when the unemployment peak was universal because the entire country was shut down. During this timeframe, many large urban counties were not included in our sample because many were rapid risers at the beginning of the pandemic. Thus, the urban category from this timeframe may not be representative of all urban areas. It would be misleading to make a conclusion on the urban/rural differences based on our current sample. We clarified the second limitation regarding representativeness of large urban areas (Lines 219-223):

“Second, because we limited the analysis to first-time rapid riser counties that occurred in a time period after the U.S. unemployment peak in April 2020, we did not assess the effects in counties that were identified as rapid risers prior to July 2020 when many large urban areas were first identified as rapid riser. Almost two-thirds of US counties are non-metropolitan and a little over fifty percent of the counties examined in our sample are non-metropolitan counties. Thus, these results may not be representative of the entire United States.”

To your second suggestion, we have made a US map with rapid riser counties shaded and used different color for counties with different quartile of overall-SVI. We added the figure into our supplementary document: S1 Fig. 

3. For the Discussion, I think it’s quite crucial to emphasize the increased need for non-English language resources in terms of unemployment benefits, employment opportunities, general healthcare, and COVID information in particular considering the impact on non-English language speakers reflected here.

Thank you for your comments! We agree that the need for non-English language resources is crucial. We added a sentence to discuss non-English speaker residents’ social needs in lines 242 to 246. Now it reads as below:

“In socially vulnerable communities disproportionately affected by COVID-19 and experiencing high rates of unemployment such as those with high proportion of racial and ethnic minority and non-English speaker residents, addressing the social needs (e.g., access to health care, food, health insurance, non-English language resources for economic benefits, health care, and accurate COVID-19 information) to help support public health measures such as testing, contact tracing, and vaccination may be needed during any infectious disease pandemic or public health crisis.”

4. I saw just a few minor typos. Please reread for copy editing.

Thank you! We have proofread it and corrected the typos. 

5. Overall an excellent study. Thank you!

Thank you for your positive comments!

---

## [Decision Letter · Decision Letter 1]

10 Mar 2022

Change in Unemployment by Social Vulnerability among United States Counties with Rapid Increases in COVID-19 Incidence — July 1–October 31, 2020

PONE-D-21-27081R1

Dear Dr. Tang,

We’re pleased to inform you that your manuscript has been judged scientifically suitable for publication and will be formally accepted for publication once it meets all outstanding technical requirements.

Kind regards,

Candace C. Nelson

Academic Editor

PLOS ONE

Additional Editor Comments (optional):

Thank you for addressing the comments and suggestions of the Reviewers. After a close review of this manuscript, I have one further request: Please remove the percentage signs ("%") from the beta values and corresponding confidence intervals listed in Table 1, Beta values can sometimes be interpreted as percentage change (which is true in this case), and it's fine to do so in the text to help the reader interpret the results, but they are not actually percentages and shouldn't be represented as such in the table.

Reviewers' comments:

Reviewer's Responses to Questions

**Comments to the Author**

1. If the authors have adequately addressed your comments raised in a previous round of review and you feel that this manuscript is now acceptable for publication, you may indicate that here to bypass the “Comments to the Author” section, enter your conflict of interest statement in the “Confidential to Editor” section, and submit your "Accept" recommendation.

Reviewer #1: All comments have been addressed

2. Is the manuscript technically sound, and do the data support the conclusions?

Reviewer #1: Yes

3. Has the statistical analysis been performed appropriately and rigorously? 

Reviewer #1: Yes

4. Have the authors made all data underlying the findings in their manuscript fully available?

Reviewer #1: Yes

5. Is the manuscript presented in an intelligible fashion and written in standard English?

Reviewer #1: Yes

6. Review Comments to the Author

Reviewer #1: Thank you for addressing the comments of the reviewers! I think the manuscript has been improved and is an important contribution to the scientific literature.

7. PLOS authors have the option to publish the peer review history of their article (what does this mean?). If published, this will include your full peer review and any attached files.

Reviewer #1: **Yes: **Emily Sparer-Fine

---

## [Editor Report · Acceptance letter]

24 Mar 2022

PONE-D-21-27081R1 

Change in unemployment by social vulnerability among United States counties with rapid increases in COVID-19 incidence — July 1–October 31, 2020    

Dear Dr. Tang:

I'm pleased to inform you that your manuscript has been deemed suitable for publication in PLOS ONE. Congratulations! Your manuscript is now with our production department. 

Kind regards, 

on behalf of

Dr. Candace C. Nelson 

Academic Editor

PLOS ONE